# Family structure and the association with physical activity—Findings from 40 countries participating in the Health Behaviour in School-aged Children (HBSC) study

Ellen Haug[1,2]*, Otto Robert Frans Smith[2,3], Kwok Ng[4,5,6,7], Oddrun Samdal[1], Adilson Marques[8,9], Alberto Borraccino[10], Jaroslava Kopcakova[11,12], Leila Oja[13], Anne-Siri Fismen[14]

1 Department of Health Promotion and Development, University of Bergen, Bergen, Norway, 2 Department of Teacher Education, NLA University College, Bergen, Norway, 3 Department of Health Promotion, Norwegian Institute of Public Health, Bergen, Norway, 4 Faculty of Education, University of Turku, Turku, Finland, 5 Physical Activity for Health Research Centre, Department of Physical Education and Sport Sciences, Health Research Institute, University of Limerick, Limerick, Ireland, 6 School of Educational Sciences and Psychology, University of Eastern Finland, Finland, 7 Faculty of Sport and Health Sciences, University of Jyvaskyla, Jyvaskyla, Finland, 8 CIPER, Faculdade de Motricidade Humana, Universidade de Lisboa, Lisbon, Portugal, 9 ISAMB, Universidade de Lisboa, Lisbon, Portugal, 10 Department of Clinical and Biological Sciences Public Health and Paediatrics, University of Torino, Orbassano (TO), Italy, 11 Department of Health Psychology and Research Methodology, Faculty of Medicine, P.J. Safarik University in Kosice, Kosice, Slovakia, 12 Olomouc University Social Health Institute, Palacky University Olomouc, Olomouc, Czech Republic, 13 National Institute for Health Development, Estonia, 14 Faculty of Health and Social Sciences, Western Norway University of Applied Sciences, Inndalsveien, Bergen, Norway

* ellen.haug@uib.no

## Abstract

### Background

The family has been acknowledged as central to developing physical activity (PA) beliefs and behaviours. However, increased diversity in family structures has developed over the last decades. This study examines the association between family structure and PA among adolescents and cross-national variations in the associations.

### Methods

The data are from the 2013/14 Health Behaviours in School-Aged Children study, involving nationally representative samples of 11-, 13- and 15-year-olds (n = 211,798) from 40 countries. Multilevel Poisson regression analysis was used to examine the associations between family structure and moderate to vigorous physical activity (MVPA) and vigorous physical activity (VPA) by age, gender, socioeconomic status (SES), and geographic region.

### Results

Living with one versus two parents was associated with a reduced likelihood of daily 60 min MVPA for boys (IRR = 0.96, 95% CI: 0.92, 0.99) and ≥ 4 times/week VPA (IRR 0.93, 95% CI: 0.91, 0.95). This impact on MVPA differed across individual-level SES (high SES; IRR =

**Data Availability Statement:** Data are available from the HBSC Institutional Data access (www.

hbscdata.uib.no) for external researchers who meet the criteria for access to the data.

**Funding:** The Slovak HBSC data collection and JK was supported by the Slovak Research and Development Support Agency under Contract no. APVV-22-0078. The funders had no role in study design, data collection and analysis, decision to publish, or preparation of the manuscript.

**Competing interests:** The authors have declared that no competing interests exist.

0.92, (p <0.05), low SES; IRR = 1.04, (ns)), and was for VPA only significant for those with siblings (IRR = 0.93, 95% CI: 0.91, 0.96). Cross-country variations in the association between living with one versus two parents were observed, most pronounced for VPA. These differences varied by region, primarily explained by country-level SES differences between regions. The likelihood of daily 60 min MVPA also increased with siblings in the main house (IRR 1.11, 95% CI: 1.07, 1.14), and ≥ 4 times/week VPA decreased with grandparents in the main house (IRR 0.91, 95% CI: 0.89,0.94).

## Conclusions

Family structure correlated with PA, but cross-country differences exist. The findings are relevant for the development of policies and programs to facilitate PA, especially in countries where living with one versus two parents was unfavourable. Additional country-specific research is needed to identify challenges for engaging in PA related to family structure.

## Introduction

Being physically active is linked to numerous health benefits, both from a medical and psychological perspective. Children and adolescents who regularly practice physical activity (PA) have a better lipid profile, less body fat, better levels of physical fitness, and a lower risk of having metabolic syndrome [1, 2]. Adolescents who are more physically active also have fewer complaints of psychosomatic symptoms [3] and are less likely to experience depression [1, 4], suicidal ideation [5], and other mental health problems [6]. In addition, adolescents who practice more PA are likely to have better academic performance [6, 7]. However, despite the benefits of PA, many children and adolescents don't comply with governmental recommendations and the PA guidelines from the World Health Organization (WHO) [8, 9]. In Europe, large within and between-country differences in PA levels are typically observed [8]. As low levels of PA during adolescence compromise present and future health, factors that may influence adolescents' PA behaviour must be identified to inform the development of evidence-based strategies and interventions [10].

Socio-ecological models emphasise that PA is a complex behaviour determined by a broad range of factors at multiple levels [11]. The family typically represents the earliest setting for PA experiences and has been acknowledged as central to developing PA beliefs and behaviours [12]. Parental influences have been of particular interest. Parents can either directly or indirectly affect adolescents' PA with supportive actions relating to encouragement, involvement, transport facilitation and economic aid [13, 14] by fostering a motivational climate [15, 16] and as role models for PA [17, 18].

However, adolescents in Westernised societies increasingly live in various family unit structures. Over the past decades, more children and adolescents are growing up in one-parent families, in joint custody or reconstructed families consisting of a parent and a stepparent [19–21]. This trend of increased diversity in family structures has led to an interest in how the family in which children grow up may affect adolescents' health and health behaviours. In general, the literature suggests that living in households other than a traditional family constituted by both parents is less favourable for various health behaviours [22–25] and also for weight status [26]. Research on the relationship between family structure and adolescent PA behaviours has so far produced mixed results. Some studies report no differences in measures of PA between

children from one versus two-parent families [27–31]. However, most studies find positive associations with two-parent families [32–38]. In contrast, fewer studies report positive associations between one-parent [39–42] or stepparent families [41]. However, a better understanding of the association between family structure and PA from the European context and with a cross-national perspective is needed.

In most previous studies, family structure has been categorised as single- vs dual-parent households, ignoring the possible impact of other family structures, such as living with a stepparent in the house, having siblings or living with grandparents. For instance, siblings are assumed to contribute to both positive and negative experiences in PA and sports through encouragement, support, jealousy, and rivalry, and with varied experiences of PA, depending on sex composition [12]. Thus, a broader examination of family structures can provide a more nuanced understanding of family composition as a contextual correlate of young people's PA.

Understanding why disparities exist according to family structure has become increasingly important. It has been suggested that family structure is related to socioeconomic status (SES) [43] and that living with one (versus two parents) may be associated with socioeconomic disadvantages [44]. This underlines the importance of considering family structure in light of social inequalities when addressing adolescents' health and health behaviours. From a socioecological perspective [11], the current study will add to the existing literature by providing an extensive cross-country examination of the relationship between family structure and PA, which opens up a broader understanding of how environmental factors, policies, and the organisation of adolescent sport, may interplay with an individual's PA across family structures.

Hence, the objective of the present study was to examine associations between various family structures and moderate to vigorous physical activity (MVPA) as well as vigorous physical activity (VPA) among adolescents across the WHO European region and Canada participating in the "Health Behaviour in School-aged Children" study. A WHO Cross-national study" (HBSC). We also examined the cross-country variations in the associations between living with one versus two parents in PA.

## Materials and methods

### Study design and data collection

The present study is based on nationally representative data from adolescents aged 11, 13, and 15 years from 40 countries participating in the HBSC survey in 2013/2014. The HBSC study aims to enhance the understanding of young people's health behaviours in their social settings. The students answered a standardised questionnaire at school after receiving instructions from their teacher. Oral and written information on the confidentiality of their responses was provided, and participation was voluntary. Most countries used school class as the primary sampling unit (some countries used schools as the sampling unit). Schools/classes that declined to participate and students absent on the day the survey was carried out were the two main sources of non-response and were not followed up. In most countries included, response rates at the school, class, or student level exceeded 80% [45].

This study was conducted according to the guidelines in the Declaration of Helsinki. The HBSC study was approved by the Regional Committees for Medical and Health Research Ethics that approved this study in Norway, with additional approvals at the country level based on national requirements for this type of study (please see HBSC_ethics_2014_supplementary file for details)." Parental written or passive informed consent to participate was obtained in accordance with requirements from the national/local ethical boards. The HBSC Data Management Centre checked the quality of the data collected, performed appropriate cleaning, and merged national data sets into an international data file. The methodology for data collection is

described in the HBSC protocol [46], which prescribes consistency in sampling plans, survey instruments and data collection. Detailed information about the study is available at http://www.hbsc.org/.

## Measures

**Demographic.** Gender was measured as either boy or girl. The participant also reported the month and year of birth, which was then calculated based on the survey completion time. After rounding to the nearest age group, they were subsequently grouped as 11-, 13- and 15-year-olds.

**Family structure.** Family structure was measured by a single item: "Please answer this first question for the home where you live all or most of the time and tick the people who live there". The response categories were mother, father, stepmother (or father's partner), stepfather (or mother's partner), grandfather, grandmother, foster home, and others. The data were coded into three categories: one parent in the main home, both parents in the main home, and no parents in the main home. Participants in the latter category (1.8%) were excluded from all analyses. Separate binary variables were derived for stepparent in the main home (yes/no) and grandparent(s) in the main home (yes/no) and were included as covariates. Having siblings in the main home (yes/no) was derived from two items referring to where the respondent lived all or most of the time: "Please indicate how many brothers and sisters live here (including half, step or foster brothers and sisters)" (How many brothers?, How many sisters?).

**Moderate to vigorous physical activity.** MVPA was measured with a single item introduced by the following definition of PA intensity levels: "Physical activity is any activity that increases your heart rate and makes you get out of breath some of the time. Examples, including local examples, were provided with the statement, "Physical activity can be done in sports, school activities, playing with friends, or walking to school" before asking the following question, "Over the past 7 days, on how many days were you physically active for a total of at least 60 minutes per day?". Please add up all the time you spent in physical activity each day" with possible responses ranging from 0 to 7 days [47]. The item has reasonable validity, moderate reliability [48, 49], and acceptable correlation with accelerometer measures [50, 51]. To reflect the daily PA recommendations, we dichotomised the item with a cut-off point for daily MVPA of at least 60 min daily.

**Vigorous physical activity.** VPA was measured by asking the respondents the following question: "Outside school hours: How often do you usually exercise in your free time so much that you get out of breath or sweat?" with the possible responses: Every day/4 to 6 times a week/2 to 3 times a week/Once a week/Once a month/Less than once a month/Never. To reflect international recommendations, the cut-off for participating in VPA regularly was set to four or more times a week following the international HBSC report [8]. The item has good reliability [52, 53]. Validity was fair when correlated with maximal oxygen consumption [53] and with accelerometer measurement [50].

**Socioeconomic status.** SES was assessed using the family affluence scale (FAS) [54]. FAS is a measure of material affluence derived from the characteristics of the family's household and consists of six items (family car, number of computers, own bedroom, family holidays, number of bathrooms, dishwasher in home). FAS is considered a valid SES indicator [55] and also for cross-national comparison [56]. Each student was assigned an individual FAS score (individual-level SES) ranging from 0 (low) to 13 (high), and each country had a mean FAS score (country-level SES), which was calculated from individual-level FAS within the respective country. The SES indicators were included in the analysis.

**Country classifications.** European subregions were coded according to the EuroVoc classification [57], encompassing four separate regions. Canada was included in the Western European group. Israel was included in the Southern European group.

## Statistics

Multilevel Poisson regression analysis was used to examine the associations between family structure and measures of MVPA and VPA. Level-1 units were students, and level-2 units were classes. All countries were pooled together for analysis, and the country variable was modelled as a fixed effect [58]. We started with a simple random intercept model with family structure as the only covariate (model 1). In the next steps, all level-1 predictors (gender, age, individual-level SES, stepparent in main home, grandparent(s) in main home, siblings in main home, and country) were first added as main effects (model 2), followed by a model that included the 2-way interactions of the model 2 predictors with family structure. Non-significant interactions based on the Wald-test were deleted and the model was re-run with significant interactions only (model 3). To examine whether potential country variations in the association between family structure on the one hand and MVPA and VPA, on the other hand, could be explained by geographical region or country-level family affluence, the country-by-family structure interaction of model 3 was replaced by the two cross-level interactions that included the mentioned country-level variables and family structure (model 4a: geographical region; model 4b: geographical region and country-level family affluence). Individual family affluence (individual-level SES) was group-mean centred and used as a level-1 predictor, whereas country-level family affluence (country-level SES) was grand-mean centred. Categorical variables were left uncentered. All analyses were conducted in STATA v.15.

## Results

The current sample included 211,798 adolescents (49.2% boys) from 40 countries. Boys were underrepresented in the Irish (38.9%) and the Russian (43.8%) samples. Table 1 reports cross-country heterogeneity in family structure, individual-level SES, country-level SES and proportion of adolescents reporting daily 60 min MVPA and ≥ 4 times/week VPA. The percentages of adolescents living with one parent in the main home ranged from 6.4 in Albania to 38.3 in Greenland. In the total sample, 23.2% lived with one parent, 8.7% with a stepparent in addition to one of their parents, and 15.6% with grandparents in the main home. Moreover, 83.2% had siblings in the main home. In total, 20.4% reported daily 60 min MVPA, ranging from 10.3% in Italy to 28.6% in Bulgaria. In comparison, 25.4% participated ≥ 4 times/week VPA in their free time, ranging from 9.4% in Armenia to 46.4% in Norway. The country-level mean SES (FAS) varied from 4.9 (Albania) to 9.9 (Luxemburg).

### Family structure differences in MVPA

As shown in Table 2 (model 1), the unadjusted analysis indicated that adolescents living with one versus two parents had a lower likelihood of daily 60 min MVPA. However, after accounting for covariates, the association was no longer statistically significant (model 2). The results of model 2 indicated that having sibling(s) in the main home (IRR = 1.11, 95% CI: 1.07, 1.14), higher individual-level SES (IRR = 1.05, 95% CI: 1.04, 1.05), and being a boy (IRR = 1.60, 95% CI: 1.57, 1.64) were associated with a greater likelihood of daily 60 min MVPA, while 13-year-olds (IRR = 0.77, 95% CI: 0.75, 0.79) and 15-year-olds (IRR = 0.61, 95% CI: 0.59, 0.63) were associated with a lower likelihood of daily MVPA than 11-year-olds.

The interaction analysis showed that the strength of the association between living with one versus two parents and MVPA was more pronounced in boys than girls (Wald test for

**Table 1. Characteristics of the study population (n = 215 509 students)\*.**

| | N | Boys (%) | 13 yr (%) | 15 yr (%) | Country Level SES | One parent in the main home (%) | Stepparent in the main home (%) | Grandparent (s) in the main home (%) | Sibling (s) in the main home (%) | Daily 60 MIN MVPA (%) | ≥ 4 times/ week VPA (%) |
|---|---|---|---|---|---|---|---|---|---|---|---|
| **Western Europe** | | | | | | | | | | | |
| Austria | 3416 | 46.5 | 31.7 | 37.0 | 9.0 | 22.5 | 7.6 | 20.0 | 89.2 | 19.9 | 33.5 |
| Belgium (French) | 5814 | 49.7 | 33.7 | 32.7 | 8.5 | 26.9 | 16.9 | 6.3 | 91.9 | 17.9 | 31.3 |
| Belgium (Flemish) | 4359 | 54.9 | 27.0 | 39.5 | 8.9 | 23.1 | 14.5 | 14.2 | 89.8 | 15.2 | 38.6 |
| Canada | 12530 | 49.5 | 37.3 | 38.5 | 8.7 | 28.5 | 11.2 | 6.1 | 86.0 | 25.0 | 34.1 |
| Germany | 5893 | 50.9 | 35.1 | 35.6 | 8.9 | 23.4 | 9.9 | 14.4 | 86.0 | 15.4 | 32.9 |
| England | 5264 | 51.9 | 29.9 | 30.4 | 8.8 | 27.6 | 11.3 | 8.1 | 90.4 | 18.3 | 22.8 |
| France | 5627 | 50.4 | 38.6 | 30.9 | 8.8 | 27.5 | 13.1 | 6.7 | 90.9 | 13.0 | 24.3 |
| Ireland | 4064 | 38.9 | 36.8 | 37.3 | 8.7 | 21.1 | 6.3 | 6.9 | 92.8 | 23.8 | 25.4 |
| Luxembourg | 3259 | 47.4 | 36.2 | 34.6 | 9.9 | 25.8 | 12.1 | 6.8 | 88.7 | 22.7 | 38.5 |
| Scotland | 5806 | 50.3 | 35.4 | 32.4 | 8.8 | 31.4 | 11.8 | 4.9 | 89.8 | 17.6 | 32.5 |
| Switzerland | 6592 | 49.5 | 36.0 | 33.9 | 9.6 | 20.2 | 8.1 | 8.3 | 90.8 | 14.6 | 38.6 |
| Wales | 5041 | 51.0 | 36.5 | 27.7 | 9.1 | 35.5 | 11.8 | 4.2 | 86.1 | 16.4 | 23.3 |
| **Eastern Europe** | | | | | | | | | | | |
| Albania | 5011 | 49.0 | 33.1 | 34.5 | 4.9 | 6.4 | 0.8 | 39.1 | 95.6 | 27.9 | 10.2 |
| Armenia | 3640 | 47.7 | 31.7 | 28.5 | 5.3 | 9.5 | 0.1 | 51.7 | 97.6 | 22.7 | 9.4 |
| Bulgaria | 4586 | 52.1 | 32.3 | 34.4 | 6.8 | 21.0 | 5.1 | 34.5 | 57.8 | 28.6 | 16.0 |
| Croatia | 5696 | 50.2 | 34.9 | 33.8 | 7.2 | 14.1 | 4.2 | 33.9 | 88.4 | 25.6 | 17.6 |
| Czech Republic | 4999 | 47.6 | 34.0 | 34.9 | 8.0 | 29.4 | 12.0 | 23.6 | 86.8 | 21.4 | 21.9 |
| Hungary | 3845 | 49.6 | 34.8 | 28.2 | 6.4 | 28.5 | 10.2 | 13.8 | 85.2 | 22.5 | 21.2 |
| Republic of Moldova | 4472 | 50.8 | 33.3 | 33.3 | 5.3 | 19.3 | 4.7 | 37.8 | 84.4 | 26.2 | 12.1 |
| North Macedonia | 4137 | 49.8 | 31.5 | 35.0 | 6.9 | 10.9 | 0.6 | 48.7 | 100.0 | 26.9 | 14.4 |
| Poland | 4475 | 49.7 | 33.6 | 32.8 | 6.9 | 20.6 | 6.6 | 23.5 | 84.3 | 24.2 | 21.2 |
| Romania | 3824 | 47.4 | 31.4 | 36.8 | 5.6 | 20.8 | 3.9 | 25.8 | 76.6 | 22.2 | 16.1 |
| Russian Federation | 4616 | 43.8 | 38.2 | 31.5 | 6.2 | 29.7 | 10.6 | 25.5 | 85.2 | 18.0 | 16.7 |
| Slovakia | 6076 | 50.3 | 40.2 | 30.6 | 7.2 | 22.4 | 0.9 | 0.8 | 86.8 | 25.1 | 19.8 |
| Slovenia | 4950 | 48.8 | 34.8 | 32.5 | 9.0 | 18.0 | 6.1 | 30.4 | 87.6 | 18.4 | 20.9 |
| Ukraine | 4466 | 47.4 | 30.5 | 36.9 | 5.3 | 25.6 | 8.4 | 34.4 | 70.1 | 26.2 | 14.9 |
| **Northern Europe** | | | | | | | | | | | |
| Denmark | 3867 | 46.8 | 35.3 | 32.9 | 9.2 | 24.2 | 10.1 | 1.2 | 93.6 | 13.2 | 41.6 |
| Estonia | 3980 | 50.3 | 35.3 | 31.1 | 7.5 | 32.1 | 14.0 | 16.5 | 84.7 | 16.6 | 23.2 |
| Finland | 5878 | 49.2 | 32.4 | 33.7 | 8.4 | 24.4 | 14.5 | 4.5 | 94.2 | 27.9 | 39.6 |
| Greenland | 927 | 47.6 | 36.8 | 30.7 | 5.8 | 38.3 | 15.2 | 8.9 | 90.8 | 17.3 | 17.2 |
| Iceland | 10490 | 50.0 | 35.3 | 31.7 | 8.5 | 27.7 | 13.5 | 2.7 | 91.1 | 22.5 | 24.9 |
| Lithuania | 5578 | 50.8 | 35.3 | 29.4 | 8.1 | 27.0 | 8.9 | 24.4 | 86.4 | 20.8 | 28.8 |
| Latvia | 5375 | 47.8 | 35.2 | 31.0 | 6.5 | 32.6 | 11.5 | 22.2 | 78.7 | 18.7 | 19.8 |
| Norway | 3380 | 48.7 | 30.8 | 28.8 | 9.8 | 19.9 | 9.8 | 4.6 | 93.1 | 18.9 | 46.4 |
| Sweden | 7515 | 49.6 | 29.6 | 36.3 | 9.2 | 28.2 | 10.7 | 2.1 | 93.2 | 14.2 | 36.0 |

*(Continued)*

**Table 1.** (Continued)

| | N | Boys (%) | 13 yr (%) | 15 yr (%) | Country Level SES | One parent in the main home (%) | Stepparent in the main home (%) | Grandparent (s) in the main home (%) | Sibling (s) in the main home (%) | Daily 60 MIN MVPA (%) | ≥ 4 times/ week VPA (%) |
|---|---|---|---|---|---|---|---|---|---|---|---|
| **Southern Europe** | | | | | | | | | | | |
| Greece | 4098 | 49.7 | 35.0 | 32.0 | 6.6 | 14.0 | 3.5 | 17.5 | 86.3 | 13.5 | 24.9 |
| Israel | 6148 | 48.6 | 30.1 | 30.1 | 7.7 | 13.3 | 3.6 | 6.0 | 100.0 | 12.4 | 23.3 |
| Italy | 4024 | 50.3 | 35.1 | 31.6 | 7.5 | 15.7 | 3.3 | 16.9 | 86.3 | 10.3 | 22.8 |
| Malta | 2214 | 51.4 | 35.8 | 28.2 | 9.2 | 11.8 | 1.4 | 2.7 | 78.4 | 18.0 | 17.7 |
| Portugal | 4910 | 47.4 | 39.8 | 27.1 | 8.5 | 24.0 | 8.8 | 14.6 | 77.3 | 15.6 | 15.7 |
| Spain | 10956 | 49.2 | 38.8 | 33.9 | 8.2 | 18.5 | 5.6 | 10.8 | - [a] | 26.1 | 23.5 |
| Total | 211798 | 49.2 | 34.6 | 33.1 | 7.9 | 23.2 | 8.7 | 15.6 | 83.2 | 20.4 | 25.4 |
| %missing | - | 0 | 0.8 | | 8.3 | 4.6 | 4.6 | 4.6 | 5.7 | 2.8 | 5.3 |

* The reference groups (girls, 11-year-olds, living with both parents) are not presented.

[a] Incomplete data on siblings.

gender interaction: $(\chi^2(1) = 4.19, p = 0.04)$). The model-derived conditional IRR for boys was statistically significant (IRR = 0.96, 95% CI: 0.92, 0.99), whereas the conditional IRR for girls was not (IRR = 1.00, 95% CI: 0.96, 1.04). The association between living with one versus two parents and MVPA also differed across individual-level SES (Wald test for SES

**Table 2. Crude and adjusted model for associations between family structure and daily 60 min MVPA, all countries.**

| | Model 1 | | Model 2 | | Model 3 | | Model 4b | |
|---|---|---|---|---|---|---|---|---|
| | IRR | 95% CI | IRR | 95% CI | IRR | 95% CI | IRR | 95% CI |
| One parent in the main home | 0.91*** | [0.89,0.93] | 0.98 | [0.96,1.01] | 0.84 | [0.67,1.05] | 1.05 | [0.99,1.11] |
| Boys | | | 1.60*** | [1.57,1.64] | 1.62*** | [1.58,1.66] | 1.62*** | [1.58,1.66] |
| 13 yr olds | | | 0.77*** | [0.75,0.79] | 0.77*** | [0.75,0.79] | 0.77*** | [0.75,0.79] |
| 15 yr olds | | | 0.61*** | [0.59,0.63] | 0.61*** | [0.59,0.62] | 0.61*** | [0.59,0.63] |
| Individual-level SES | | | 1.05*** | [1.04,1.05] | 1.05*** | [1.05,1.06] | 1.05*** | [1.05,1.06] |
| Stepparent in the main home | | | 0.99 | [0.95,1.03] | 0.99 | [0.95,1.03] | 0.99 | [0.96,1.03] |
| Grandparent (s) in the main home | | | 1.02 | [0.99,1.04] | 1.02 | [0.99,1.04] | 1.02 | [0.99,1.04] |
| Siblings(s) in the main home | | | 1.11*** | [1.07,1.14] | 1.11*** | [1.07,1.14] | 1.11*** | [1.07,1.14] |
| One parent x boys | | | | | 0.95* | [0.91,1.00] | 0.95* | [0.91,1.00] |
| One parent x Individual-level SES | | | | | 0.99* | [0.98,1.00] | 0.99* | [0.98,1.00] |
| One parent x Country-level SES | | | | | | | 0.99 | [0.96,1.02] |
| One parent x Eastern Europe | | | | | | | 0.95 | [0.87,1.04] |
| One parent x Northern Europe | | | | | | | 0.93* | [0.87,0.99] |
| One parent x Southern Europe | | | | | | | 0.96 | [0.88,1.06] |
| Constant | 0.27*** | [0.25,0.29] | 0.24*** | [0.22,0.26] | 0.24*** | [0.22,0.26] | 0.24*** | [0.22,0.25] |
| Variance estimates | | | | | | | | |
| Random intercept | 0.08*** | [0.07,0.09] | 0.03*** | [0.02,0.04] | 0.03*** | [0.02,0.04] | 0.03*** | [0.02,0.04] |

Note: Reference categories: gender; girls, age; 11-year-olds, family structure; both parents in the main home. Country fixed effects are not shown for models 1–4. Model 4a not presented. Country x family structure (one parent) interaction is not shown for model 3.

*** p<0.001,

**p<0.01,

*p<0.05

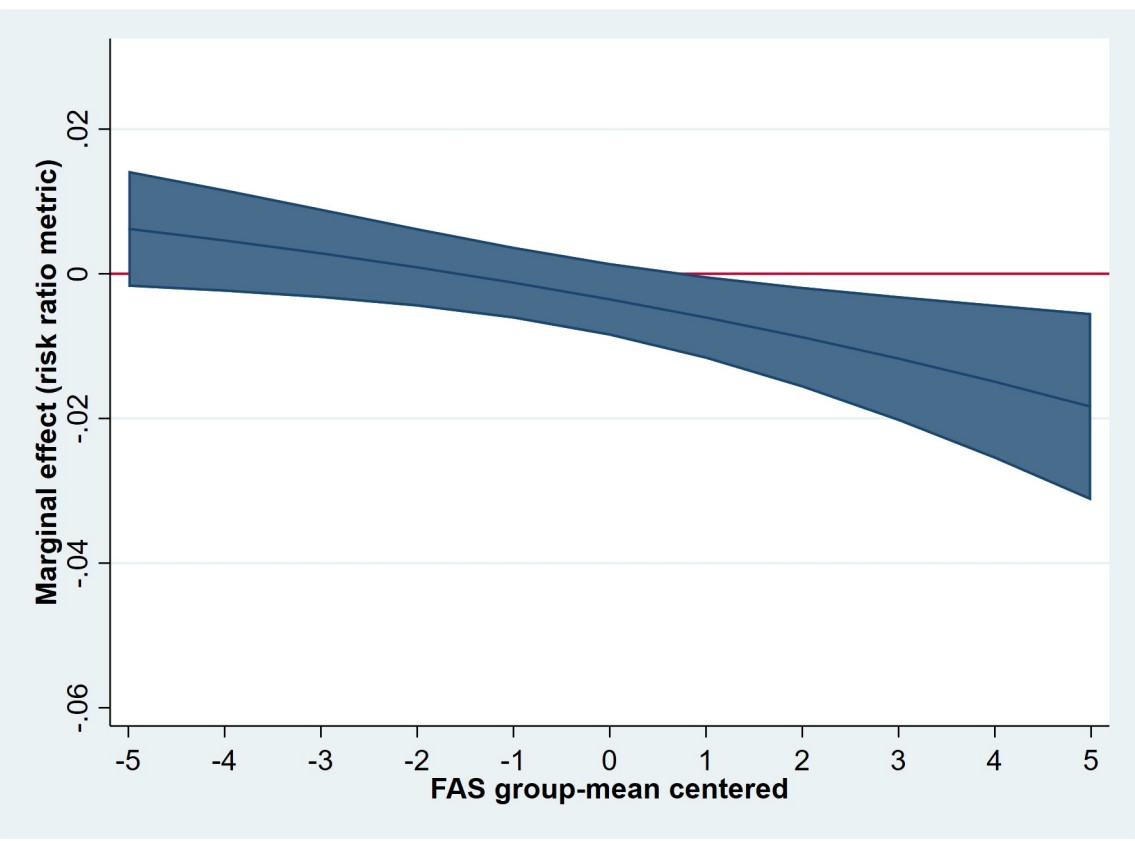

**Fig 1. Conditional marginal effects of living in one versus two parents on daily 60 min MVPA by individual-level SES (FAS) (estimates derived from model 3).**

interaction: ($\chi^2(1) = 5.85$ $p = .02$)). As shown in Fig 1, the conditional effect was most pronounced for adolescents with higher SES scores. For adolescents with a group-mean centred SES score of +5, the conditional IRR was 0.92 ($p < .05$). In contrast, the IRR for adolescents with group-mean centred SES score of -5 (low) was 1.04 (ns). The 2-way interactions between family structure (one parent) and age ($\chi^2(2) = 0.31$, $p = 0.86$), having a stepparent in the main home ($\chi^2(1) = 0.53$, $p = 0.47$), having grandparent(s) in the main home ($\chi^2(1) = 0.00$, $p = 0.98$), and having a sibling(s) in the main home ($\chi^2(1) = 0.63$, $p = 0.43$) were all not statistically significant and were therefore not included in models 3 and 4.

## Cross-national differences in MVPA

The interaction between family structure and country was also statistically significant $\chi^2(40) = 58.01$, p = 0.03) and included in model 3 (not shown in Table 2). However, when analysing country by country, the association was no longer statistically significant in the majority of countries (Fig 2). Living with one parent was only associated with a lower likelihood of MVPA in Norway, Bulgaria and Iceland (estimate points to the left of the red line), whereas living with one parent was associated with a higher likelihood of MVPA in French Belgium. The cross-national variation in the relationship between one-parent families and MVPA could not be explained by region (Wald-test: $\chi^2(3) = 4.97$, p = 0.17) (model 4a, not shown in Table 2) or country-level SES (Table 2).

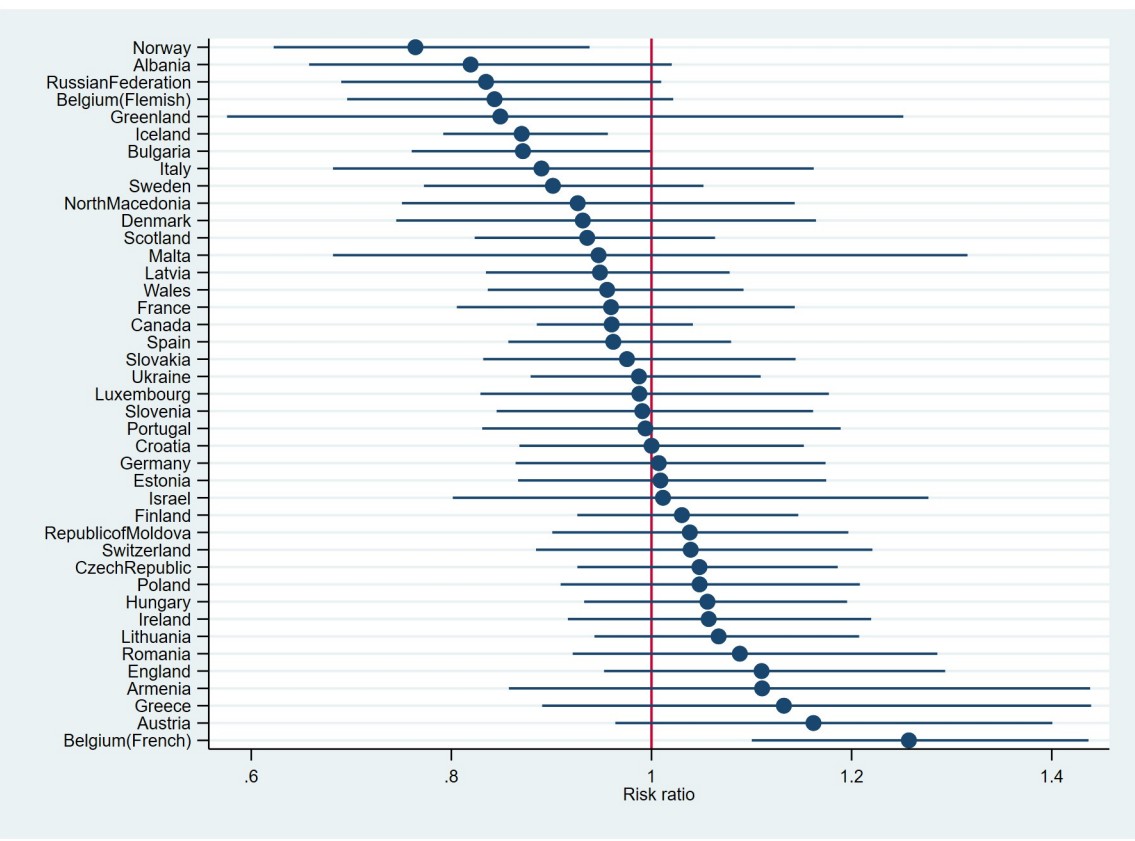

**Fig 2. Associations between living with one versus two parents and 60 min MVPA across countries, adjusted for co-variates (estimates were derived from model 3).**

## Family structure differences in VPA

As shown in Table 3, the adjusted analysis (model 2) indicated that living with one parent in the main home (IRR = 0.93, 95% CI: 0.91,0.95) and grandparents in the main home (IRR = 0.91, 95% CI: 0.89,0.94) were both associated with a lower likelihood of > 4 times/week VPA, while higher individual-level SES (IRR = 1.10, 95% CI: 1.09, 1.10), being a boy (IRR = 1.49, 95% CI: 1.46,1.51), age 13 (IRR = 1.13, 95% CI: 1.11, 1.16) and age 15 (IRR = 1.10, 95% CI: 1.07, 1.12) were associated with a higher IRR for VPA.

The interaction analysis showed that the strength of the association between living with one versus two parents and VPA was more pronounced for adolescents with sibling(s) than those without sibling(s) (Wald test for siblings interaction: ($\chi^2(1)$ = 4.19, $p$ = 0.04)). The model-derived conditional IRR for those with sibling(s) was statistically significant (IRR = 0.93, 95% CI: 0.91, 0.96), whereas the conditional IRR for those without sibling(s) was not (IRR = 1.00, 95% CI: 0.94, 1.05). There were no significant interactions between living with one parent in the main home and gender ($\chi2(1)$ = 0.34, p = 0.56), individual-level SES ($\chi2(1)$ = 0.21, p = 0.65), age ($\chi2(2)$ = 4.03, p = 0.13), living with a stepparent in the main home ($\chi2(1)$ = 3.18, p = .07), and living with grandparent(s) in the main home ($\chi2(1)$ = .2.28, p = .13). These interaction terms were therefore not included in subsequent models.

**Table 3. Crude and adjusted model for associations between family structure and VPA ≥ 4 times/week, all countries.**

| | Model 1 | | Model 2 | | Model 3 | | Model 4b | |
|---|---|---|---|---|---|---|---|---|
| | IRR | 95% CI | IRR | 95% CI | IRR | 95% CI | IRR | 95% CI |
| One parent in the main home | 0.85*** | [0.84,0.87] | 0.93*** | [0.91,0.95] | 1.04 | [0.71,1.51] | 0.99 | [0.93,1.05] |
| Boys | | | 1.49*** | [1.46,1.51] | 1.49*** | [1.46,1.51] | 1.49*** | [1.46,1.51] |
| 13 yr olds | | | 1.13*** | [1.11,1.16] | 1.13*** | [1.11,1.16] | 1.13*** | [1.11,1.16] |
| 15 yr olds | | | 1.10*** | [1.07,1.12] | 1.10*** | [1.07,1.12] | 1.10*** | [1.07,1.12] |
| Individual-level SES | | | 1.10*** | [1.09,1.10] | 1.10*** | [1.09,1.10] | 1.10*** | [1.09,1.10] |
| Stepparent in the main home | | | 0.97 | [0.94,1.00] | 0.97 | [0.94,1.00] | 0.97 | [0.94,1.00] |
| Grandparent(s) in the main home | | | 0.91*** | [0.89,0.94] | 0.91*** | [0.89,0.94] | 0.91*** | [0.89,0.94] |
| Sibling(s) in the main home | | | 1.01 | [0.98,1.03] | 1.03 | [1.00,1.06] | 1.02 | [0.99,1.05] |
| One parent x sibling(s) in the main home | | | | | 0.94* | [0.89,0.99] | 0.95 | [0.90,1.01] |
| One parent x Country-level SES | | | | | | | 0.96*** | [0.93,0.98] |
| One parent x Eastern Europe | | | | | | | 0.99 | [0.92,1.07] |
| One parent x Northern Europe | | | | | | | 0.99 | [0.94,1.04] |
| One parent x Southern Europe | | | | | | | 0.97 | [0.89,1.05] |
| Constant | 0.10*** | [0.09,0.11] | 0.08*** | [0.07,0.09] | 0.08*** | [0.07,0.09] | 0.08*** | [0.07,0.09] |
| Variance estimates | | | | | | | | |
| Random intercept | 0.03*** | [0.02,0.04] | 0.00 | [-0.01,0.01] | 0.00*** | [0.00,0.00] | 0.00 | [-0.01,0.01] |

Note: Reference categories: gender; girls, age; 11-year-olds, family structure; both parents in the main home. Country fixed effects are not shown for models 1–4.

Country x One parent interaction is not shown for model 3.

*** p<0.001,

**p<0.01,

*p<0.05

## Cross-national differences in VPA

The interaction between living with one parent in the main home and country was statistically significant (Model 3, not shown in Table 3, Wald test: $\chi^2(40) = 84.238$, $p<0.001$)). As shown in Fig 3, living with one versus two parents was associated with a lower IRR for > 4 times/week VPA in fourteen countries, including all countries in the Northern European region, except Denmark, whereas living with one parent was associated with a higher IRR for > 4 times/week VPA in the Russian Federation only. The association between living in a one versus a two-parent family and VPA was not statistically significant for the remaining countries.

The cross-national variation in the relationship between VPA and one- versus two-parent families could partly be explained by geographical region (Wald-test: $\chi^2(3) = 9.16.55$, $p = 0.03$), with a significantly different relationship between one versus two parents and VPA observed in Eastern Europe as compared to Western Europe (reference category, model 4a, not shown in Table 3). The model-derived conditional IRR of one parent was statistically significant for Western Europe (IRR = 0.93, 95% CI: 0.89,0.96) and Northern Europe (IRR = 0.94, 95% CI: 0.90,0.98) but not for Eastern Europe (IRR = 1.01, 95% CI: 0.96,1.06) and Southern Europe (IRR = 0.94, 95% CI: 0.88,1.01). After adding country-level SES to the model, the interaction between family structure (one versus two parents) and the geographical region became non-significant, suggesting that this association could largely be explained by differences in country-level SES between regions (IRR = 0.96, 95% CI: 0.93, 0.98), model 4b, see Table 3). As shown in Fig 4, the association between living with one parent and VPA was most pronounced in countries with higher mean SES scores. For example, the conditional IRR in countries with

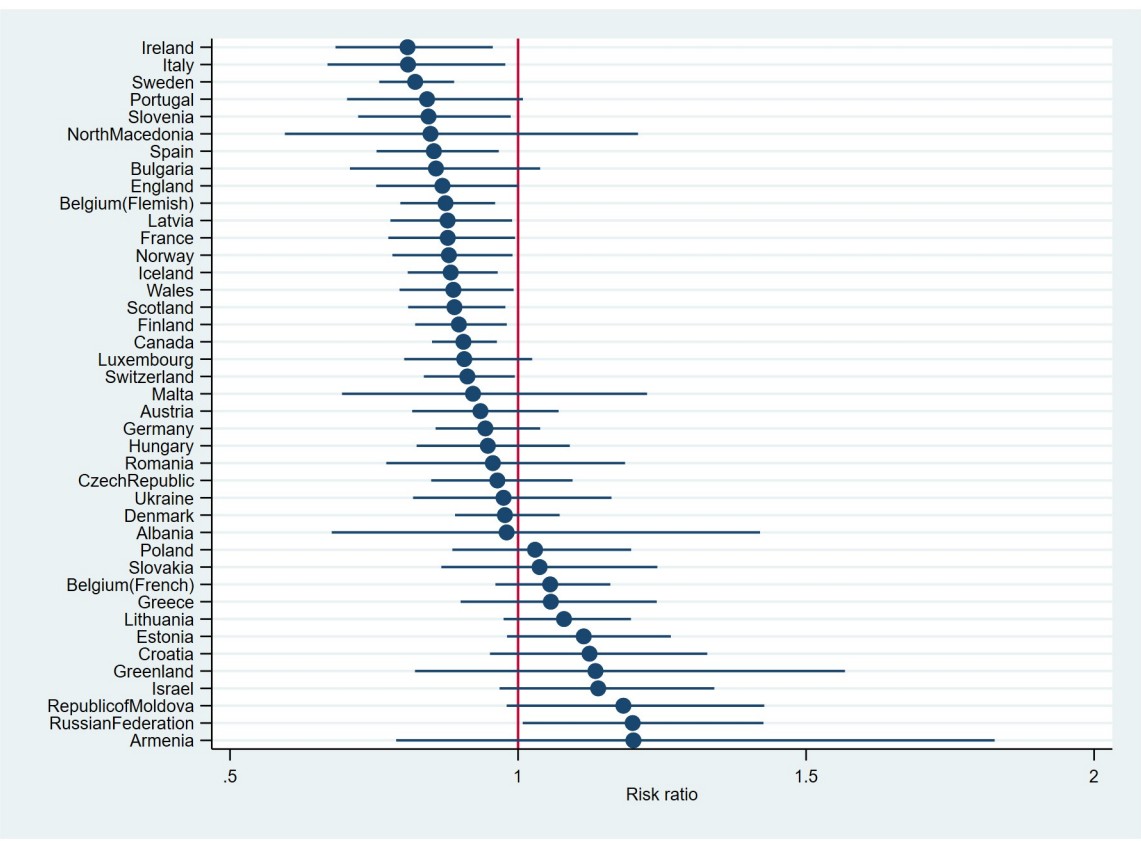

**Fig 3. Associations between living with one parent versus two parents and ≥ 4 times/week VPA across countries, adjusted for covariates.**

lower country-level SES (-2) was 1.04 (ns), whereas the conditional IRR was 0.86 ($p<0.05$) in countries with high country-level SES (+2).

## Discussion

This is the first large-scale cross-country study examining the associations between family structure and adolescent PA. Through pooling data from 40 European countries and Canada, significant associations were observed for some family structures and daily 60 min MVPA as well as ≥ 4 times/week VPA. Cross-country variations in the relationship between living with one versus two parents and PA were notable, most pronounced for VPA. These findings were related to geographical regions and largely explained by country-level SES.

### Associations between family structure and PA

In the pooled analyses, a differential gender effect was observed, with a lower likelihood for daily 60 min MVPA only for boys living with one parent compared to two parents in the main home. Previous studies have, to a limited extent, examined gender differences in the relationship between family structure and indicators of PA. In an English study, boys, but not girls living with a single parent compared to two parents, spent more time on sedentary behaviours in general and on screen time on weekdays and weekends [30]. In accordance with the displacement hypothesis, sedentary activities may replace time spent on PA [59]. Thus, increased

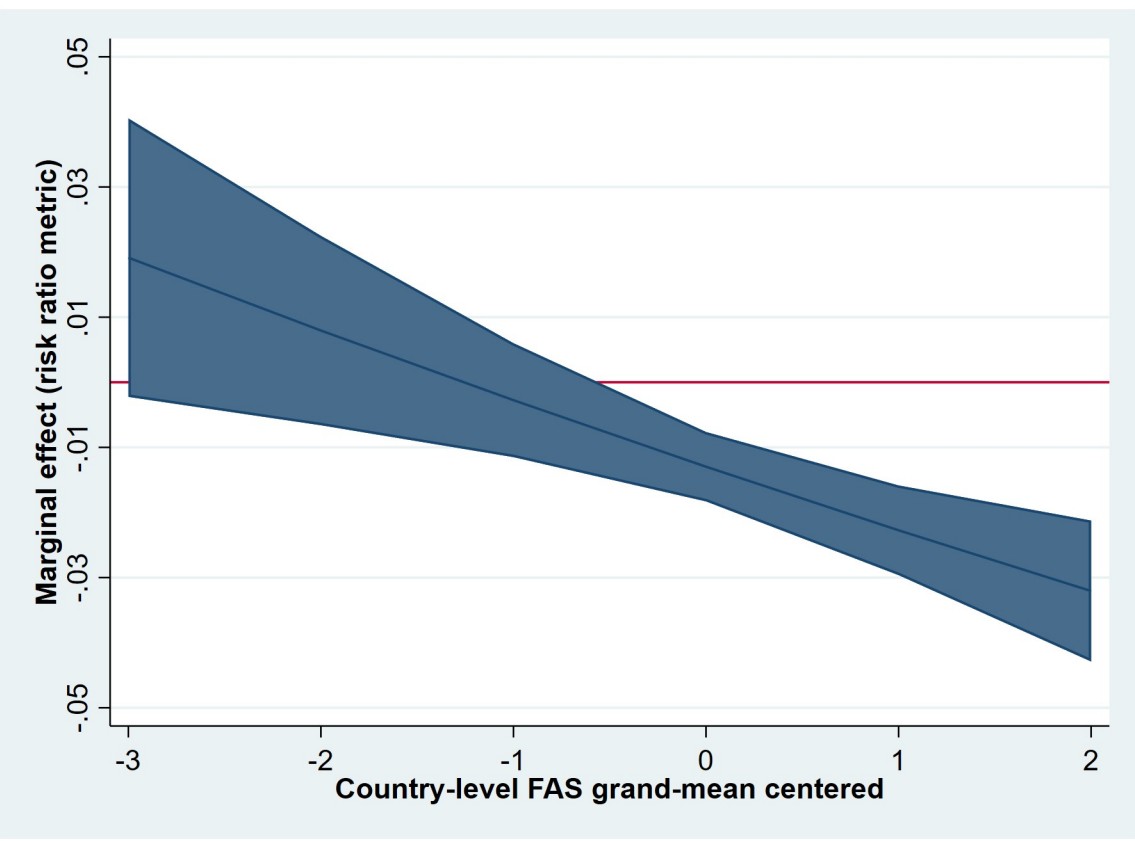

**Fig 4. Conditional marginal effects of living with one versus two parents on $\geq$ 4 times/week VPA by country-level SES (FAS) (estimates derived from model 4b).**

screen use among boys living with one parent could be one possible explanation for the current study findings that should be further examined.

Another reason for a lower likelihood of daily 60 min MVPA for boys living with one compared to two parents may relate to more boys than girls participating in sport [37, 60, 61]. Also, studies suggest that boys accrue more MVPA minutes during training than girls [62]. If sport participation constitutes a larger part of boys' overall MVPA, a reduced capacity to follow up and support sport involvement in one-parent compared to two-parent families could impact their MVPA. The finding of a lower likelihood of $\geq$ 4 times/week VPA for adolescents living with one versus two parents in the current study may suggest so. Engaging more often in leisure time VPA reflects frequent and regular involvement in PA, typically for organised sport activities that demand greater parental support. Children in one-parent families have reported experiencing more barriers to engaging in physical activities due to less parental support caused by a lack of free time, workload, and household responsibilities [63]. This can lead to fewer opportunities to support the children in sport, such as transporting them to and from leisure time activities, cheering during competitions, and doing PA with them [64]. Lower levels of sport involvement in one-parent compared to two-parent households align with studies from countries with various social and socioeconomic contexts [32–38]. Another possibility could be centred on children with two parents who live with only one or mainly one, as extra effort is needed to maintain participation in organised sport [32]. Time to travel to visit

biological parents has been reported as a factor restricting engagement in PA among children in non-traditional families [63].

Additionally, a factor that has been hypothesised to impact children's PA engagement is the family sport culture, described as the family's fundamental role in establishing deeply embodied motivations, habits, and lifestyles [65]. Strandbu and colleagues (2020) demonstrate the sustained importance of family sport culture for adolescents' involvement in sport over time [65]. One could assume that those living with one parent would have reduced chances of experiencing a sporting culture at home compared to those living with two parents. Single parents may also have less time to be physically active role models.

In the current study, the association between family structure and MVPA differed across individual-level SES, with the conditional effect of family structure (living with one versus two parents) most pronounced for those with higher SES scores. Being affluent typically strengthens the ability to cover financial outlay (e.g. sport equipment, membership, fees), and indeed, high individual-level SES was a predictor for MVPA and VPA in the current study. Nevertheless, not having the benefit of a two-parent family structure to contribute to additional support for an active lifestyle may increase the differences and explain the more substantial effect of living with two compared to one parent on MVPA, especially within the high SES group. An interaction effect of individual-level SES on VPA was, however, not observed.

Interestingly, there were no beneficial effects on MVPA and VPA of having a stepparent in the main home. This may seem surprising as stepparents could contribute with time, economic resources, and sport culture by role-modelling PA habits. Similar findings were reported for organised sport involvement among Canadian [32] and Norwegian [38] adolescents. The reasons are likely to be complex. It may be explained by underlying processes in reconstructed families and the role the family climate may play [66]. For example, studies have indicated that stepparents are, in general, less committed to their non-biological children [41, 67], reducing the potential parent-child dyad strength in support for PA of the adolescent [68]. Programmes that target improving relations between stepparents and their non-biological children may reverse this trend. Another aspect is that a reconstructed family may have difficulties and need time to form relationships between the stepparent and stepchild that facilitate PA [69].

Although the entire family unit is regarded as especially important in endorsing PA behaviours [13], research has primarily addressed the role of the parents only. However, sibling relations are reciprocal and dynamic and can contribute to PA through peer modelling, encouraging active transport, the opportunity for playmates and serving as additional caregivers [12, 70]. In our pooled analysis, having siblings in the main home increased the likelihood of MVPA compared to not having a sibling(s). This aligns with a recent meta-analysis that found children with siblings having higher PA levels, as measured by accelerometer or pedometer [70]. A direct effect of having sibling(s) in the main house was not observed for leisure time VPA for at least 4 timers/week. However, the strength of the association between living with one versus two parents and VPA was more pronounced for adolescents with siblings than those without siblings, with a significant negative impact on VPA of having siblings in the main home observed. This finding somewhat nuances the existing literature where a positive relationship has been found between having a sibling and children's participation in sport [12]. It should be mentioned that our outcome measure, $\geq 4$ times/week VPA, reflects a relatively high level of PA engagement that may be less feasible for adolescents in one-parent families with siblings, particularly if it demands substantial parental support regarding time and logistics.

The focus on grandparents' influence on obesity-related health behaviours has been mainly on children's diet behaviour and weight, with varying results across ethnicities and countries [71]. In the current study, living with grandparent(s) in the main house was unrelated to

MVPA, and negatively associated with VPA. These findings partly contrast the results from a former HBSC survey of a US sample of Latino school-children that found a higher OR of daily 60 min MVPA but no association with VPA among those living with grandparents in the house [72].

## Cross-country variations

The association between living with one versus two parents and PA varied across countries. Still, it was not statistically significant in most countries. The likelihood of daily 60 min MVPA among adolescents living with one parent was only significantly lower in three countries, whereas it was higher in one country. Thus, for most countries, factors other than family structure explain the multifaceted MVPA behaviour, supporting the need for a systems approach to promoting PA [73]. The association between living with one versus two parents and VPA varied somewhat across countries, with adolescents in fourteen countries having a lower likelihood of VPA ≥ 4 times/week. The differential associations could partly be explained by region, with a lower likelihood of living with one parent observed in Northern and Eastern Europe. The interaction between family structure and the geographical region became non-significant when controlling for country-level SES, suggesting that differences at country-level SES could largely explain this association.

A similar finding was observed when examining the impact of living with one versus two parents on overweight and obesity in a recent study based on the same sample, with stronger associations detected in the Northern/Western region that, to a large extent, was explained by country-level SES [26]. The current study findings may contribute to explaining the observed influences of one versus two parents on overweight and obesity, as PA is consistently negatively associated with overweight/obesity [1, 74]. Although countries with higher country-level SES, like those in the Northern region, tend to have more well-developed family policies and welfare systems, the situation when it comes to financial strain and poverty is still unfavourable for one-parent families, partly because their employment is more likely part-time and based on temporary contracts [44]. A study by Badura and colleagues (2021) found that adolescents from lower SES families and non-nuclear families (consisting of two parents and their child) were less likely to participate in organised activities across nine countries from Western, Central and Northern Europe and Canada with divergent social and socioeconomic contexts [37].

There may also be differences between and within the geographical regions in how sport and PA are organised that co-occur with higher country-level SES. In many countries, organised competitive sports for children and youth is carried out in the context of the school or in combination with club sport. In Nordic countries, however, voluntary competitive sports for children and youth is mostly organised outside the school system [75]. Also, much of the club sports organised for young people in these countries depend somewhat on parental involvement, e.g. as a coach or other voluntary work [38, 75]. Lack of time, a more likely challenge in one-parent families, may thus influence the ability to take on such responsibilities and become a barrier to club sport involvement.

In contrast, in Portugal, the sport system is structurally separated into club or school sport [76]. The school sport system where the students can attend freely is divided into internal activities with recreational and competitive sports inside the school and external activities that aim to specialise students in a particular sport and compete against other schools [76]. Still, significant cross-country variations in the extent and nature of school sport systems in Europe have been documented [77]. For instance, in some countries (e.g. France, Poland and Sweden), a participation fee for extracurricular school sport activities is required [77]. In Sweden, low participation rates in extracurricular activities are also explained by a negative attitude towards

competition within Swedish schools and easy access to clubs outside of school [78]. Thus, an extracurricular school sport system may reduce differences related to family structure, as it puts less demand on families to facilitate and support PA engagement. Another factor that may account for some cross-country variations is the proximity to attractive arenas for PA and sports for youth. For instance, the Nordic region is the least densely populated in Europe [79], and with sparse settlement, young people will more often depend on their parents for transportation to training facilities.

## Implications

Policymakers, the sport sector, and health professionals should know that adolescents' family context can be complex and potentially affect their involvement in PA, especially leisure time VPA. The present findings suggest that the existing welfare policies do not necessarily eliminate family structure differences. The role of both family support and the structural organisation of youth sports should be addressed in public PA initiatives, and policy actions should support families with limited time and poorer access to PA facilities. Also, as the proportion of children and adolescents living in ONE-parent or stepparent families continues to grow, it is important to monitor PA habits by family structure, together with overall population trends. Lastly, the current study should also be considered from a broader public health perspective, as unfavourable PA levels add to several other negative health behaviours observed among adolescents living in ONE-parent or stepparent families, for example, dietary behaviours, smoking, and substance use [23–25].

## Strengths and limitations

There are several strengths and limitations of the study that should be considered. The large sample size gave sufficient statistical power to examine interaction effects related to family structure. The study also controlled for several important covariates. A strength is the use of well-established measures and comparable individual-level data from 40 countries based on comprehensive methodological data collection procedures.

However, the study has some limitations. All data were self-reported, known to have recall and reporting bias [80]. However, most items have been documented to have satisfying validity and reliability [46]. Of note, SES was measured by FAS, an indicator of material affluence [54]. The associations between SES and family structure differences may differ for other SES indicators, as FAS in a Swedish study was only moderately correlated with parental income and weakly correlated with parents' occupational status [81].

In this study, we lack information on how long the participants lived in their current family structure, the PA motivational climate in the family [16], and other household characteristics (e.g., BMI and PA of the primary caretaker), limiting our ability to study the processes involved. limiting our ability to study the processes involved. In addition, we could not differentiate between those living in a one-parent family with no involvement from the other parent or those part-time or nearly 50/50 with each parent with potentially extensive involvement of both parents. The influence of the school environment facilitation on the amount of PA, which is independent of parental SES, was also not considered. Finally, the study had a cross-sectional design and unobserved sources of the observed association were not accounted for, which makes it difficult to propose any causality.

## Conclusion

As the structure of families is changing in many countries, studying the associations between family structure and PA is important at this time. Pooled data demonstrated that family

structure correlated with daily 60 min MVPA and $\geq$ 4 times/week VPA. Still, cross-country differences in associations between living with one versus two parents and PA highlight the value of collecting comparable cross-national data on adolescent health behaviours and their social contexts. The study findings should be considered in developing policies and programs that aim to facilitate PA, especially in countries where living with one versus two parents was associated with a lower likelihood of PA. More country-specific research is needed to address different types of psychosocial challenges and stress relevant to one-parent families in particular.

## Acknowledgments

HBSC is an international study carried out in collaboration with the World Health Organization/Europe WHO/EURO. The authors acknowledge the International coordinator of the 2013/2014 survey, Professor Candace Currie and her team at the University of St. Andrews, Scotland, and the International Data Centre Manager, Professor Oddrun Samdal and her team at the University of Bergen, Norway. We also thank all the participating students, staff and schools that took part in the HBSC survey, as well as all PIs from the countries examined.

## Author Contributions

**Conceptualization:** Ellen Haug, Otto Robert Frans Smith, Anne-Siri Fismen.

**Data curation:** Ellen Haug, Oddrun Samdal.

**Formal analysis:** Otto Robert Frans Smith.

**Investigation:** Ellen Haug.

**Methodology:** Ellen Haug.

**Writing – original draft:** Ellen Haug.

**Writing – review & editing:** Otto Robert Frans Smith, Kwok Ng, Oddrun Samdal, Adilson Marques, Alberto Borraccino, Jaroslava Kopcakova, Leila Oja, Anne-Siri Fismen.

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
