## [Decision Letter · Decision Letter 0]

12 Oct 2023

PONE-D-23-23392Family Structure and the association with physical activity – findings from 40 Countries participating in the Health Behaviour in School-aged Children (HBSC) StudyPLOS ONE

Dear Dr. Haug,

Thank you for submitting your manuscript to PLOS ONE. After careful consideration, we feel that it has merit but does not fully meet PLOS ONE’s publication criteria as it currently stands. Therefore, we invite you to submit a revised version of the manuscript that addresses the points raised during the review process.

ACADEMIC EDITOR: Dear Author,There are still corrections that need to be made in this manuscript. Please attend to all the reviewers' comments. 

We look forward to receiving your revised manuscript.

Kind regards,

Zulkarnain Jaafar

Academic Editor

PLOS ONE

Journal Requirements:

Reviewers' comments:

Reviewer's Responses to Questions

**Comments to the Author**

1. Is the manuscript technically sound, and do the data support the conclusions?

Reviewer #1: Yes

Reviewer #2: Partly

2. Has the statistical analysis been performed appropriately and rigorously? 

Reviewer #1: I Don't Know

Reviewer #2: No

3. Have the authors made all data underlying the findings in their manuscript fully available?

Reviewer #1: Yes

Reviewer #2: No

4. Is the manuscript presented in an intelligible fashion and written in standard English?

Reviewer #1: Yes

Reviewer #2: Yes

5. Review Comments to the Author

Reviewer #1: Dear Authors,

this is a well-written manuscript that needs only a few minor changes.

There are several places in the paper where you could use the abbreviation for physical activity (PA) instead of the full name.

Figure 2 and Figure 3 are a bit fuzzy and should be adjusted.

Please state the novelty and practical implications of your study.

Reviewer #2: Thank you for the opportunity to review this manuscript on family structure associated with physical activity in school-aged children. Please find my comments below.

My major concerns are as follows: (1) Not all factors associated with PA were controlled for; for example, caregiver’s BMI and PA could largely affect child’s BMI and PA, which is not accounted for in this paper; (2) a family structure could signal other issues in the family, leading to omitted variable bias on the family structure variable; (3) the countries in the sample are extremely varied and have very different SES levels and community supports for family and PA (including built environment, sports facilities, school curriculum) – it is possible that fixed effects are too limited to account for these differences; (4) ORs in a model with a non-rare outcome over-inflates the relative risk and should not be used.

More detailed comments are below.

I wonder if variance decomposition could be used to see how different family structures contribute to the variation in PA (compared to other variables).

Line 149: If having parent + stepparent was included as a separate binary covariate in the regression, the model assumes that one can have one parent and parent+stepparent at the same time (which is impossible). I wonder if “parent + step parent” should be included as a fourth category in the main covariate (one parent, 2 parents, parent + step parent, no parents). This could also be done in a separate model.

Line 170: Would it be possible to treat MVPA and VPA as a continuous variable as a supplemental analysis? Dichotomizing a variable leads to some loss of information, and ORs often represent an inflated measure of risk.

Line 180: Odds ratios (as mentioned above) represent an inflated measure of risk. While logit models are appropriate for use with binary variables, odds ratios can only be used to approximate relative risk when the outcome is rare. In the descriptive statistics, the outcome ranges from 9% to ~30%, making this not a rare outcome. In this case, authors should estimate relative risk instead of using odds ratios. Given that authors used Stata, it is very straightforward to obtain measures of both absolute difference in risk and relative risk from any model, including logit. Please refer to the following publications for methods:

Kleinman, Lawrence C., and Edward C. Norton. "What's the risk? A simple approach for estimating adjusted risk measures from nonlinear models including logistic regression." Health services research 44.1 (2009): 288-302.

Norton, Edward C., Morgen M. Miller, and Lawrence C. Kleinman. "Computing adjusted risk ratios and risk differences in Stata." The Stata Journal 13.3 (2013): 492-509.

Line 198: I wonder if the authors have attempted to test other ways of controlling for country differences. The sample includes extremely different countries with different supports for families and PA. It would make sense to test for both fixed and random effects, as well as looking at certain regions separately. I would like to see some evidence that a fixed effects model is the best model for this type of sample.

Line 195: Why wasn’t the country-level SES used in the models? What about other household characteristics (such as age, gender, BMI, PA of primary caretaker, BMI of the child, etc)?

Line 195: It would be helpful to clearly list all models with all of their covariates in each model; otherwise, it is hard to understand what covariates were included and how – from the Statistics paragraph as written. Perhaps include this as a supplemental table if there is no space in the text.

6. PLOS authors have the option to publish the peer review history of their article (what does this mean?). If published, this will include your full peer review and any attached files.

Reviewer #1: No

Reviewer #2: No

---

## [Author Response · Author response to Decision Letter 0]

7 Dec 2023

Dear ACADEMIC EDITOR: 

We want to thank the reviewers for their constructive and positive feedback about the manuscript, which we believe has improved the quality of our paper. We have addressed all comments and provided a point-by-point response. To clarify, reviewers’ comments are in bold font, and, where appropriate, we have inserted the revised text in blue, with new text highlighted. Please see our rebuttal letter and updated manuscript.

---

## [Decision Letter · Decision Letter 1]

23 Feb 2024

Family Structure and the association with physical activity – findings from 40 Countries participating in the Health Behaviour in School-aged Children (HBSC) Study

PONE-D-23-23392R1

Dear Dr.Melingen Haug,

We’re pleased to inform you that your manuscript has been judged scientifically suitable for publication and will be formally accepted for publication once it meets all outstanding technical requirements.

Kind regards,

Zulkarnain Jaafar

Academic Editor

PLOS ONE

Additional Editor Comments (optional):

Reviewers' comments:

Reviewer's Responses to Questions

**Comments to the Author**

1. If the authors have adequately addressed your comments raised in a previous round of review and you feel that this manuscript is now acceptable for publication, you may indicate that here to bypass the “Comments to the Author” section, enter your conflict of interest statement in the “Confidential to Editor” section, and submit your "Accept" recommendation.

Reviewer #3: All comments have been addressed

2. Is the manuscript technically sound, and do the data support the conclusions?

Reviewer #3: Yes

3. Has the statistical analysis been performed appropriately and rigorously? 

Reviewer #3: I Don't Know

4. Have the authors made all data underlying the findings in their manuscript fully available?

Reviewer #3: Yes

5. Is the manuscript presented in an intelligible fashion and written in standard English?

Reviewer #3: Yes

6. Review Comments to the Author

Reviewer #3: The topic of this research article is interesting. and I can see that the authors have made an effort to address all the comments of previous authors.

7. PLOS authors have the option to publish the peer review history of their article (what does this mean?). If published, this will include your full peer review and any attached files.

Reviewer #3: No

---

## [Editor Report · Acceptance letter]

4 Apr 2024

PONE-D-23-23392R1 

PLOS ONE

Dear Dr. Haug, 

I'm pleased to inform you that your manuscript has been deemed suitable for publication in PLOS ONE. Congratulations! Your manuscript is now being handed over to our production team.

Kind regards, 

on behalf of

Dr. Zulkarnain Jaafar 

Academic Editor

PLOS ONE